# BAG OF TRICKS FOR FGSM ADVERSARIAL TRAINING

## ABSTRACT

Adversarial training (AT) with samples generated by Fast Gradient Sign Method (FGSM), also known as FGSM-AT, is a computationally simple method to train robust networks. However, during its training procedure, an unstable mode of "catastrophic overfitting" has been identified in (Wong et al., 2020), where the robust accuracy abruptly drops to zero within a single training step. Existing methods use gradient regularizers or random initialization tricks to attenuate this issue, whereas they either take high computational cost or lead to lower robust accuracy. In this work, we provide the first study, which thoroughly examines a collection of tricks from three perspectives: *Data Initialization, Network Structure, and Optimization*, to overcome the catastrophic overfitting in FGSM-AT.

Surprisingly, we find that simple tricks, *i.e.*, a) masking partial pixels (even without randomness), b) setting a large convolution stride and smooth activation functions, or c) regularizing the weights of the first convolutional layer, can effectively tackle the overfitting issue. Extensive results on a range of network architectures validate the effectiveness of each proposed trick, and the combinations of tricks are also investigated. For example, trained with PreActResNet-18 on CIFAR-10, our method attains 49.8% accuracy against PGD-50 attacker and 46.4% accuracy against AutoAttack, demonstrating that pure FGSM-AT is capable of enabling robust learners.

## 1 INTRODUCTION

Convolution neural networks (CNNs), though achieving compelling performances on various visual recognition tasks, are vulnerable to adversarial perturbations (Szegedy et al., 2014). To effectively defend against such malicious attacks, adversarial examples are utilized as training data for enhancing model robustness, a process known as adversarial training (AT). To generate adversarial examples, one of the leading approaches is to perturb the data using the sign of the image gradients, namely the Fast Gradient Sign Method (FGSM) (Goodfellow et al., 2015).

The adversarial training with FGSM (FGSM-AT) is computationally efficient, and it lays the foundation for many followups (Kurakin et al., 2017; Tramèr et al., 2018; Madry et al., 2018; Xie et al., 2019; Zhang et al., 2019). Nonetheless, interestingly, FGSM-AT is not widely used today because of the catastrophic overfitting: the model robustness will collapse after a few training epochs (Wong et al., 2020). Several methods are proposed to mitigate catastrophic overfitting and stabilize FGSM-AT. For instance, (Wong et al., 2020) pre-add uniformly random noises around images to generate adversarial examples, *i.e.*, turning the FGSM attacker into the PGD-1 attacker. (Andriushchenko & Flammarion, 2020) propose GradAlign, which regularizes the AT via explicitly maximizing the gradient alignment of the perturbations. While these approaches successfully alleviate the catastrophic overfitting, there are still some limitations. For example, GradAlign requires an extra forward pass compared to the vanilla FGSM-AT, which significantly increases the computational cost; Fast-AT in (Wong et al., 2020) shows relatively lower robustness, and may still collapse when used to train larger networks or applied in the larger-perturbation settings.

In this paper, we aim to develop more effective and computationally efficient solutions for attenuating catastrophic overfitting. Specifically, we revisit FGSM-AT and design to stabilize its training from the following three perspectives:

- **Data Initialization.** Following the idea of adding random perturbations in (Madry et al., 2018; Wong et al., 2020), we propose to randomly mask a subset of the input pixels to stabilize FGSM-AT,

dubbed FGSM-Mask. Surprisingly, additional analysis suggests that the randomness of the masking process may not be necessary during training—we find that applying a pre-defined masking pattern to the training set also effectively stabilizes FGSM-AT. This observation also holds for adding perturbations as the attack initialization in (Wong et al., 2020), challenging the general belief that randomness is one of the key factors for stabilizing AT.

- **Network Structure.** We identify two architectural elements that affect FGSM-AT. Firstly, in addition to boosting robustness as shown in (Xie et al., 2020), we find that a smoother activation function can make FGSM-AT more stable. Secondly, we find vanilla FGSM-AT can effectively train Vision Transformers (ViTs) (Dosovitskiy et al., 2021) without showing catastrophic overfitting. We conjecture this phenomenon may be related to how CNNs and ViTs extract features: *i.e.*, CNNs typically extract features from overlapped image regions (*i.e.*, stride size < kernel size in convolution), while ViTs extract features from non-overlapped image patches (*i.e.*, stride size = kernel size in convolution). By simply increasing the stride size of the first convolution layer in a CNN, we validate that the resulting model can stably train with FGSM-AT.

- **Optimization.** Inspired by GradAlign (Andriushchenko & Flammarion, 2020), we propose ConvNorm, a regularization term that simply constrains the weights of the first convolution layer to stabilize FGSM-AT. Different from GradAlign which introduces a significant amount of extra computations, our ConvNorm works as nearly computationally efficiently as the vanilla FGSM-AT.

**Our contributions.** In summary, we discover a bag of tricks that effectively alleviate the catastrophic overfitting in FGSM-AT from three different perspectives. We extensively validate the effectiveness of our methods with a range of different network structures on the popular CIFAR-10/100 datasets, using different perturbation radii. Our results demonstrate that only using FGSM-AT is capable of enabling robust learners. We hope this work can encourage future exploration on unleashing the potential of FGSM-AT.

## 2 PRELIMINARIES

Given a neural classifier $f$ with parameters $\theta$, we denote $x$ and $y$ as the input data and the corresponding label from the data generator $D$, respectively. $\delta$ represents the adversarial perturbation, $\epsilon$ is the maximum perturbation size under the $l_\infty$-norm constraint, and $\mathcal{L}$ is the cross-entropy loss typically used for image classification tasks.

**Adversarial Training:** (Madry et al., 2018) formulates the adversarial training as a min-max optimization problem:

$$\min_\theta \mathbb{E}_{(x,y)\sim D}\Big[\max_{\|\delta\|_\infty \leq \epsilon} \mathcal{L}(f_\theta(x + \delta), y)\Big]. \tag{1}$$

Among different attacks for generating adversarial examples, we chose two popular ones to study:

- **FGSM:** (Goodfellow et al., 2015) first proposes Fast Gradient Sign Method (FGSM) to generate the perturbation $\delta$ in a single step, as the following:

$$\delta = \epsilon \, \text{sign}(\nabla_x \mathcal{L}(f_\theta(x), y)), \tag{2}$$

- **PGD:** (Madry et al., 2018) proposes a strong iterative version with a random start based on FGSM, named Projected Gradient Descent (PGD):

$$x_{t+1} = \Pi_{\|\delta\|_\infty \leq \epsilon}\left(x_t + \alpha\text{sign}(\nabla_{x_t}\mathcal{L}(f_\theta(x_t), y))\right), \tag{3}$$

where $\alpha$ denotes the step size of each iteration, $x_t$ denotes the adversarial examples after t steps, and $\Pi_{\|\delta\|_\infty \leq \epsilon}$ refers to the projection to the $\epsilon - Ball$. Compared to FGSM, PGD provides a better choice for generating adversarial examples, but it will also be much more computationally expensive. In the following sections, we call adversarial training with FGSM as FGSM-AT, and correspondingly, with PGD as PGD-AT.

**Catastrophic Overfitting:** (Wong et al., 2020) argues that non-zero initialization for perturbations is the key to avoiding the overfitting issue in adversarial training, and proposes to add uniformly random noises around clean images as the attack initialization. The detailed procedure is illustrated in the following equations:

$$\eta = \mathrm{Uniform}(-\epsilon, +\epsilon),$$
$$\delta = \delta + \alpha \, \mathrm{sign}(\nabla_x \mathcal{L}(f_\theta(x + \eta), y)), \tag{4}$$
$$\delta = \max(\min(\delta, \epsilon), -\epsilon).$$

The later work (Andriushchenko & Flammarion, 2020) alternatively proposes GradAlign to stabilize FGSM-AT, which maximizes the gradient alignment between various sets as:

$$\mathbb{E}_{(x,y)\sim D}\left[1 - \cos\left(\nabla_x \mathcal{L}(f_\theta(x), y), \nabla_x \mathcal{L}(f_\theta(x + \eta), y)\right)\right]. \tag{5}$$

## 3 BAG OF TRICKS

We aim to investigate simple yet effective solutions to overcome catastrophic overfitting in FGSM-AT. Specifically, our methods are developed from three general perspectives: *Data Initialization, Network Structure, and Optimization*. We extensively test these methods on the popular CIFAR-10 dataset (Krizhevsky & Hinton, 2009) with PreActResNet-18 (He et al., 2016). We set the maximum perturbation size $\epsilon = 8/255$ under the $\ell_\infty$-norm constraint. Unlike (Rebuffi et al., 2021; Gowal et al., 2021; Sehwag et al., 2022), the training data comes exclusively from CIFAR-10 and no extra data is used. Two adversarial attacks are considered for comprehensively evaluating model robustness: a) PGD-50 attacker (Madry et al., 2018) with 10 random restarts (PGD-50-10), where we apply untargeted mode using the ground-truth annotations and set the step size $\alpha = 2/255$; and b) the AutoAttack (AA) suite (Croce & Hein, 2020b), which includes Auto-PGD-CE, Auto-PGD-Targeted, FAB (Croce & Hein, 2020a), and Square attack (Andriushchenko et al., 2020).

**Adversarial training setups.** We set the training framework and hyper-parameters following (Pang et al., 2021). We apply an SGD optimizer with a momentum of 0.9, weight decay of $5 \times 10^{-4}$, and an initial learning rate of 0.1. We do not apply label smoothing and set ReLU as the default activation function. We apply random flip and random crop to pre-process training data, and train all models for a total of 110 epochs. The learning rate decays by $10\times$ at $100^{th}$ epoch and $105^{th}$ epoch, respectively. We use the last checkpoint to run robustness evaluations 3 times independently, and report results in the format of *mean ± var*. All of our experiments are conducted with NVIDIA TITAN XP GPUs.

### 3.1 DATA INITIALIZATION

(Wong et al., 2020) first identifies the catastrophic overfitting phenomenon in FGSM-AT and proposes to solve it by adding uniformly random noise around images as attack initialization, namely Fast FGSM-AT (F+FGSM). Though this method has shown the capability to prevent general catastrophic overfitting, it attains relatively lower robustness (*i.e.*, see the second row in Table 1). Moreover, the later work (Andriushchenko & Flammarion, 2020) argues that Fast FGSM-AT may still collapse when used to train larger networks or applied in the larger-perturbation setting.

| Methods | Clean | PGD-50-10 | AA |
|---|---|---|---|
| PGD-10 | 82.6±0.15% | 51.9±0.07% | 48.2±0.06% |
| F+FGSM | 86.3±0.02% | 45.4±0.03% | 41.0±0.03% |
| FGSM-Mask | 82.4±0.01% | 48.5±0.01% | 44.2±0.00% |
| FGSM-Mask-fixed | 80.7±0.01% | 47.2±0.02% | 43.0±0.02% |

Table 1: Robustness of different data initialization methods combined with FGSM-AT.

**FGSM-Mask.** Inspired by the core idea of F+FGSM, in this paper, we first propose to *mask a random subset of the input pixels* to stabilize the training procedure of FGSM-AT, which we term as FGSM-Mask. Figure 1 illustrates the differences between FGSM-Mask and F+FGSM when generating adversarial examples. Specifically, at each iteration, FGSM-Mask zeros out some randomly chosen pixels of each image $x$ with a mask $M$ according to a given mask ratio; then the masked image $x \otimes M$ is fed to the model to generate adversarial examples via FGSM as:

$$\delta = \alpha \, \mathrm{sign}(\nabla_{x \otimes M} \mathcal{L}(f_\theta(x \otimes M), y)), \tag{6}$$

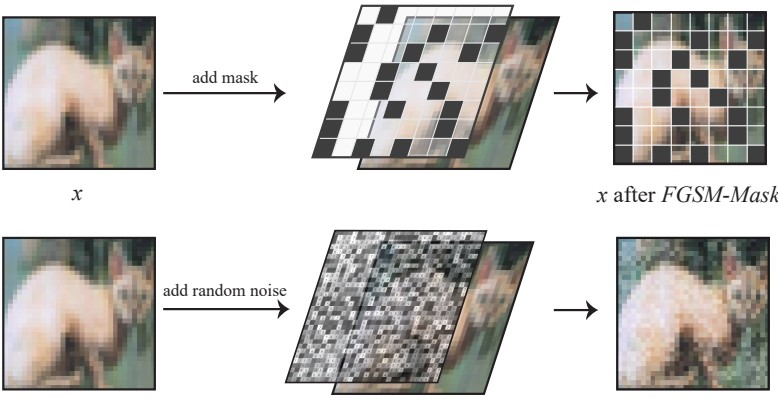

Figure 1: The illustration of the differences between FGSM-Mask and F+FGSM on initializing images for attacking.

Compared with the random noise initialization method in F+FGSM (*i.e.*, Equation equation 4), our method presents a much simpler formulation—FGSM-Mask simply randomizes the *mask* instead of manipulating the original pixel values.

To demonstrate the effectiveness of FGSM-Mask, we mask images with different ratios and report the corresponding robust accuracy in Table 2 and Figure 2 (a). Firstly, with a mask ratio of 0%, FGSM-Mask degenerates to the vanilla FGSM-AT, and therefore it suffers from catastrophic overfitting. As the mask ratio increases, the models trained with FGSM-Mask become more stable. For example, a small mask ratio like 10% or 20% can already largely mitigate the overfitting issue; the robustness only collapses at nearly the end of training. Furthering the mask ratio to 30% or above can completely resolve this issue: the robust accuracy remains stable and smooth during the training process. Specifically, we note a mask ratio of 30% competitively leads to 48.5% robust accuracy against PGD-50-10 and 44.2% robust accuracy against AA, outperforming F+FGSM (45.4% in PGD-50-10 and 41.0% in AA) by more than 3% shown in Table 1.

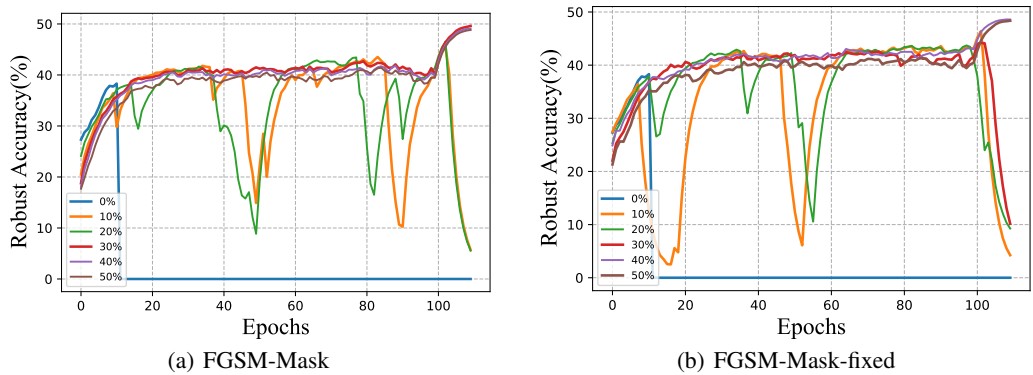

(a) FGSM-Mask          (b) FGSM-Mask-fixed

Figure 2: Robust accuracy of FGSM with various mask ratios ranging from 0% to 50%. (a) is with the random mask, and (b) is with the fixed mask.

**FGSM-Mask-Fixed.** Additionally, we observe a surprising result that the randomness of masking is not even necessary—instead, simply using a fixed masking pattern per image throughout the whole training process is enough to help stabilize FGSM-AT. In other words, we could pre-define a masked dataset offline that naturally enables the stable

| Randomized Mask Ratio | PGD-50-10 | Fixed Mask Ratio | PGD-50-10 |
|---|---|---|---|
| 0~20% | 0.0% | 0.0~20% | 0.0% |
| 30% | 48.5% | 30% | 0.0% |
| 40% | 47.9% | 40% | 47.2% |
| 50% | 47.6% | 50% | 47.2% |

Table 2: Robust accuracy of FGSM-Mask and FGSM-Mask-Fixed, when applied with different mask ratios.

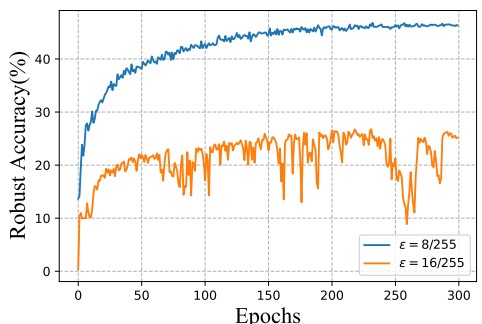
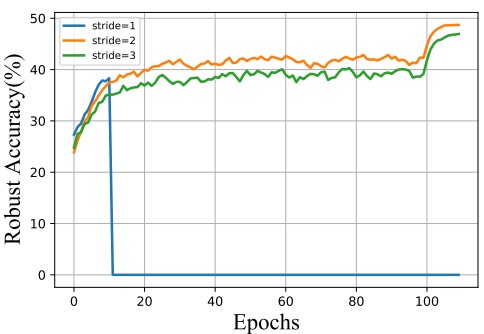

Figure 3: Robust accuracy of CVT with different maximum perturbation size $\epsilon = \{8, 16\}$.

Figure 4: Robust accuracy of CNN with different stride sizes in the first convolution layer.

training of FGSM-AT. We call this method FGSM-Mask-Fixed. As shown in Table 2 and Figure 2 (b), models trained with FGSM-Mask-Fixed achieve strong robust accuracy without applying any additional tricks. For example, with a mask ratio of 50%, the model trained with FGSM-Mask-Fixed attains a robust accuracy of 47.2% against PGD-50-10, outperforming the F+FGSM baseline by about 2%.

This observation further inspires us to revisit the random noise initialization strategy in F+FGSM. Following FGSM-Mask-Fixed, we directly apply a fixed noise pattern per image to the whole dataset. Interestingly, we note that such a strategy is also capable of helping FGSM-AT prevent catastrophic overfitting, achieving a robust accuracy of 45.2% against PGD-50-10. This finding challenges the previous belief that the randomness of initialization at each training iteration plays a vital role in ensuring the success of AT (Wong et al., 2020; Chen et al., 2022)

## 3.2 NETWORK STRUCTURE

Existing studies demonstrate that a well-designed network structure can improve model robustness (Xie et al., 2019; 2020; Singla et al., 2021; Wu et al., 2021; Guo et al., 2020; Berger et al., 2022; Tang et al., 2021). We hereby are interested in the newly emerged ViT architecture, which shows better potentials than CNN in robustness (Bai et al., 2021; Paul & Chen, 2022; Shao et al., 2021). We choose Compact Vision Transformer (CVT) (Hassani et al., 2021) as the specific instantiation of ViT architecture, and check whether CVT will encounter catastrophic overfitting in FGSM-AT.

Figure 3 shows robust accuracy during the training process. We can observe that, without any additional tricks, CVT smoothly gains robustness along the training. This conclusion still holds when training with a much more challenging adversary (*i.e.*, increasing the maximal perturbation size $\epsilon$ from 8 to 16): though we see a bigger variation in the robust accuracy along the training, CVT still successfully avoids catastrophic overfitting and attains a non-trivial final performance. These results motivate us to explore whether traditional CNN architectures can be improved to address the catastrophic overfitting problem in FGSM-AT, an aspect largely overlooked in previous studies. More specifically, we plan to borrow existing modules from ViT to help CNNs tackle catastrophic overfitting in FGSM-AT, which we detail next.

| Methods | Clean | PGD-50-10 | AA |
|---|---|---|---|
| PGD-10 | 82.6±0.15% | 51.9±0.07% | 48.2±0.06% |
| FGSM-Str2 | 83.1±0.02% | 47.2±0.05% | 44.4±0.03% |
| FGSM-Smooth | 74.9±0.09% | 47.0±0.07% | 43.0±0.03% |

Table 3: Robustness of different neural architectural modifications combined with FGSM-AT.

**Larger stride for the first convolution layer.** One big difference between ViTs and CNNs is how they process image inputs. ViTs begin with a patchify operation, which first splits an image into a sequence of non-overlapping patches, and then projects each patch with a trainable linear layer. This is usually implemented by a convolution with a large stride (*e.g.* 4 for CVT on CIFAR-10). Whereas CNNs usually adopt a much more mild downsampling strategy. Take PreActResNet-18 as an example: its first layer is a $3 \times 3$ convolutional layer with a stride of 1. To mimic ViTs, we

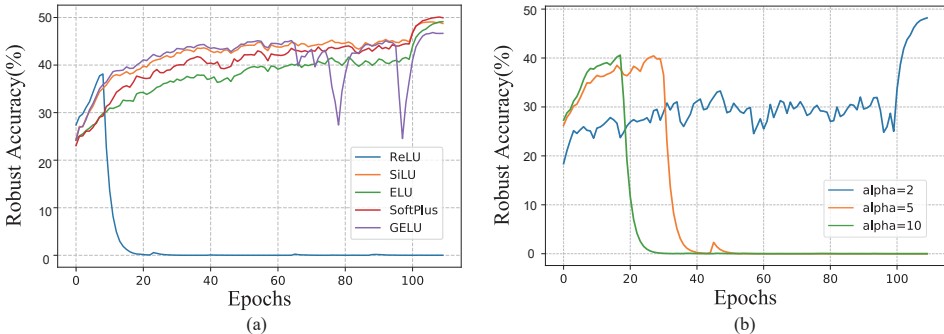

Figure 5: (a) shows robust accuracy with various activation functions. (b) shows the robust accuracy under Softplus with different $\alpha$.

propose to enlarge the stride of the first convolution layer of CNNs[1]. Interestingly, we note that the catastrophic overfitting problem is alleviated by simply increasing the stride size from 1 to 2 or 3. As shown in Figure 4, when the stride is set to 1, the robust accuracy quickly drops to zero; but when the stride is set to 2 or 3, the robust accuracy becomes more stable along the training. Among different stride options in our study, we find that FGSM-AT with a stride of 2 achieves the highest robust accuracy, *i.e.*, 47.2% against PGD-50-10 as shown in Table 3. We name this method FGSM-Str2.

**Smooth activation function.** Another important difference between ViTs and CNNs is the activation function. The common choice of CNNs' activation function is ReLU, while ViTs typically adopt a smoother activation function, GELU (Hendrycks & Gimpel, 2016). Previous studies have shown the effectiveness of smooth activation function in boosting model robustness (Xie et al., 2020; Singla et al., 2021; Gowal et al., 2020), but none of them studies it from the perspective of tackling catastrophic overfitting, which we aim to study here. Specifically, we experiment with replacing ReLU with four different smooth activation functions: SiLU (Ramachandran et al., 2018), ELU (Clevert et al., 2016), SoftPlus (Nair & Hinton, 2010), and GELU (Hendrycks & Gimpel, 2016). We show their robust accuracy along FGSM-AT in Figure 5(a) .

Firstly, we can observe that all four activation functions largely mitigate or even fully prevent catastrophic overfitting. More interestingly, we notice the degree of smoothness affects the robustness. For instance, ELU is smoother than GELU, and accordingly, the robust accuracy of ELU is stabler than that of GELU along the training. Following (Xie et al., 2020), we choose Parametric SoftPlus to systematically study the effect of function smoothness, by adjusting the scalar $\alpha$:

$$f(\alpha, x) = \frac{1}{\alpha} \log(1 + \exp(\alpha x)). \tag{7}$$

Figure 5(b) shows the robust accuracy of Parametric SoftPlus with varying $\alpha$ along the training. With a smaller value of $\alpha$, we can validate that the activation function becomes smoother and the robust accuracy becomes stabler along the training. We choose Parametric SoftPlus with $\alpha = 2$ as our baseline shown in Table 3, as it performs the best among smooth activation functions. We call this method FGSM-Smooth.

### 3.3 OPTIMIZATION

Previous works show that adding an extra regularization term can effectively prevent catastrophic overfitting in FGSM-AT. However, these methods usually introduce extra computation overhead. One typical example is GradAlign (Andriushchenko & Flammarion, 2020), which significantly increase training cost as an extra forward and backward propagation is required to compute the gradient of an adversarial set $\nabla_x \mathcal{L}(f_\theta(x + \eta), y)$ (see Equation equation 5). We hereby introduce two alternative approaches to stabilize FGSM-AT: a) directly regularizing the $L_2$ norm of gradients on input images, referred to as *GradNorm*; and b) regularizing the weights on the first layer, referred to as *WeightNorm*.

---

[1]As a larger stride will lead to a smaller feature map, we meanwhile decrease the stride of one intermediate convolution layer to keep the size of the final feature map stay the same.

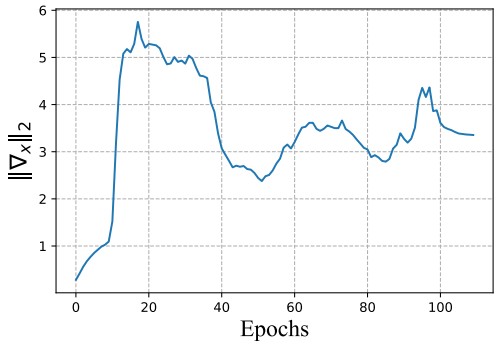

Figure 6: Trend of $\|\nabla_x\|_2$ along training. It abruptly increases when overfitting happens.

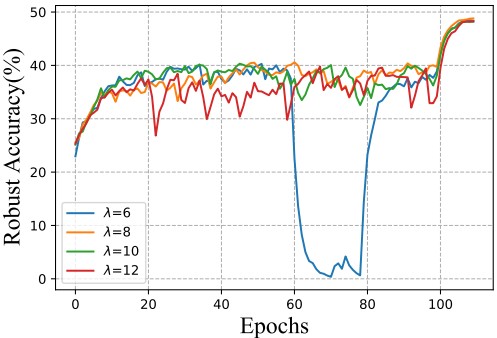

Figure 7: Robust accuracy *w.r.t.* different $\lambda$ in the proposed WeightNorm (see Equation 9).

| Methods | Clean | PGD-50-10 | AA |
|---|---|---|---|
| PGD-10 | 82.6±0.15% | 51.9±0.07% | 48.2±0.06% |
| FGSM+GradAlign | 81.2±0.12% | 47.6±0.14% | 44.0±0.06% |
| FGSM+GradNorm | 82.3±0.03% | 45.9±0.04% | 42.7±0.04% |
| FGSM+WeightNorm | 81.7±0.02% | 46.9±0.00% | 42.8±0.01% |

Table 4: Robustness of different optimization methods combined with FGSM-AT.

**GradNorm.** By taking a closer look at the $L_2$ norm of gradients $\|\nabla_x\|_2$ on input images, we observe that it abruptly increases by $\sim6\times$ at the $11^{th}$ epoch (*i.e.*, catastrophic overfitting happens) as shown in Figure 6. This observation aligns with the conclusion in (Kim et al., 2021), which points out that the increasing gradient norm leads to decision boundary distortion and a highly curved loss surface during adversarial training. This distortion makes the adversarially trained model vulnerable to multi-step adversarial attacks (*e.g.*, PGD attacks). Inspired by this phenomenon, we propose a new regularizer that directly constrains the gradient norm $\mathbb{E}[\|\nabla_x\|_2]$:

$$\mathcal{L} = \mathcal{L}(f_\theta(x + \delta), y) + \beta\|\nabla_x\|_2 \tag{8}$$

where $\beta$ controls regularizing strength. As shown in Table 4, GradNorm successfully overcomes overfitting and achieves a high robust accuracy of $45.9\%$ against PGD-50-10, which is comparable to GradAlign ($47.6\%$). Nonetheless, a drawback of GradNorm is that it requires even more computations than GradAlign. For example, with our hardware setup for one-epoch training, GradAlign takes 56s, while GradNorm nearly doubles the cost to 109s. The main reason is that regularizing the gradient norm involves second-order backpropagation, which is computationally expensive.

**WeightNorm.** Both GradAlign and GradNorm are highly effective in addressing the overfitting issue. However, they both suffer from high computational costs due to the additional backpropagation requirement. To address the overfitting issue and meanwhile without significantly introducing extra computations, we propose to directly enforce the intermediate feature difference between adversarial examples and clean samples to be small, as

$$\min \lambda\mathcal{L}^1(f(x), f(x + \delta)), \tag{9}$$

where the $\lambda$ controls the weight of the regularizer and $\delta$ is the adversarial perturbation.

Note that Adversarial Logits Pairing (ALP) (Kannan et al., 2018) can be regarded as a special example of WeightNorm, where the regularized intermediate feature is the logits $f_{\text{logits}}$. We hereby take another simple instantiation: we regularize the intermediate features generated by the first convolution layer $f_{\omega_1}$, where $\omega_1$ denotes the weights of the first convolution layer. Then $f_{\omega_1}(x + \delta) - f_{\omega_1}(x)$ can be re-written as $\omega_1(x + \delta) - \omega_1 x$, which is equal to $\omega_1\delta$. As regularizing $\delta$ involves the computationally expensive second-order backpropagation, we therefore treat $\delta$ here as a constant and only to regularize $\omega_1$. Experiment result in Table 4 show that WeightNorm can prevent catastrophic overfitting. Moreover, WeightNorm (42s/epoch) introduce very little extra computations to FGSM-AT (40s/epoch), and runs $\sim25\%$ faster than GradAlign. It is also worth mentioning that WeightNorm is robust to the choice of the hyperparameter $\lambda$ in Equation equation 9. For example, by ranging $\lambda$ between 6 and 12, WeightNorm always reliably prevents catastrophic overfitting, as shown in Figure 7

### 3.4 COMPONENTS COMBINATION

As shown in previous experiments, each proposed approach alone can mitigate the catastrophic overfitting problem. We next explore their possible synergy and report the results in Table 5. Firstly, we observe that all these combinations can successfully prevent overfitting. Specifically, three combinations, *Mask + Smooth*, *Str2 + Smooth*, and *Str2 + Smooth + WeightNorm*, yield the most effective solutions, *i.e.*, they all report 49.6% or stronger robust accuracy against PGD-50-10 and 46.0% or stronger robust accuracy against AA. This observation also corroborates the conclusion in (Xie et al., 2020) that the smooth activation function can substantially strengthen adversarial training. But meanwhile, we note, for the combinations like *Mask + Str2* or *Mask + WeightNorm*, they do not show further improvements. This observation suggests that approaches like Masking can only help prevent catastrophic overfitting. As a side note, given that the GradNorm (which incur heavy computations) is excluded here, all these combinations can run as fast as (or even faster than) the vanilla FGSM-AT.

| Methods | | | | | | Performances | | |
|---|---|---|---|---|---|---|---|---|
| Mask | Mask-Fixed | Str2 | Smooth | GradNorm | WeightNorm | Clean | PGD-50-10 | AA |
| ✓ | | ✓ | | | | 82.4±0.04% | 47.3±0.09% | 44.4±0.10% |
| ✓ | | | ✓ | | | 81.1±0.06% | 49.7±0.0.5% | 46.1±0.02% |
| ✓ | | | | | ✓ | 81.7±0.04% | 46.9±0.03% | 42.8±0.02% |
| | | ✓ | ✓ | | | 82.1±0.04% | 49.8±0.07% | 46.4±0.02% |
| | | ✓ | ✓ | | ✓ | 81.2±0.03% | 49.6±0.09% | 46.0±0.05% |

Table 5: Performances of FGSM-AT with combined tricks.

## 4 ABLATIONS

**Large networks.** Compared with small networks, larger networks are more likely to get overfitted to the training data, making preventing catastrophic overfitting more challenging. For example, as shown in Table 6, when increasing the network size from PreActResNet-18 to WideResNet-34-10, F+FGSM now fails to secure its robustness, *i.e.*, attaining 0% robust accuracy against PGD-50 or AA. This motivates us to re-validate the effectiveness of our proposed methods, using the large WideResNet-34-10.

| Methods | AT | Clean | PGD-50-10 | AA |
|---|---|---|---|---|
| Baseline | F+FGSM | 89.4% | 0% | 0% |
| | FGSM+GradAlign | 85.4% | 51.2% | 46.8% |
| | PGD-10 | 86.1% | 55.2% | 52.2% |
| Data initializatoin | FGSM-Mask | 80.0% | 34.4% | 31.6% |
| | FGSM-Mask-fixed | 71.8% | 23.5% | 21.0% |
| Network Structure | FGSM-Smooth | 75.2% | 46.9% | 44.1% |
| | FGSM-Str2 | 85.1% | 47.0% | 46.0% |
| Optimization | FGSM+GradNorm | 82.6% | 49.2% | 46.1% |
| | FGGSM+WeightNorm | 84.6% | 46.8% | 44.8% |

Table 6: Robust accuracy of different methods combined with FGSM-AT, using the large WideResNet-34-10. Note for methods in data initialization, we need to set a large masking ratio (*i.e.*, 70% in our experiments) to keep FGSM-AT stable.

We report the results in Table 6. Firstly, we note that all proposed methods can successfully prevent overfitting, *i.e.*, none of them gets zero robust accuracy. Nonetheless, we note that the effectiveness of different methods varies a lot. For example, while for methods like *Str2* help WideResNet-34-10 secures a competitive robustness (*e.g.*, 46.0% against AA), *FGSM-Mask* or *FGSM-Mask-Fixed* only attains 20%~30% robust accuracy against AA. Next, similar to the results in Table 5, we note that combining methods can help WideResNet-34-10 attain higher robustness. For example, by adopting both the smooth activation function and a large stride size in FGSM-AT, WideResNet-34-10 achieves 50.4% robust accuracy against PGD-50-10 and 47.3% robust accuracy against AA.

**Larger perturbation.** As mentioned in (Andriushchenko & Flammarion, 2020), a large perturbation will nullify the defense built by F+FGSM. We hereby aggressively increase the maximal perturbation size $\epsilon$ from 2 to 16 to validate the effectiveness of our proposed methods. As shown in Figure 8, most of our methods can still attain non-trivial robustness against PGD-50-10, demonstrating that they can reliably address the catastrophic overfitting when facing larger perturbations.

**CIFAR-100 dataset.** The main experiments of this paper focus on robustness in CIFAR-10. We hereby validate whether our methods can stabilize FGSM-AT on CIFAR-100 as well. As shown in Table 7, while the vanilla FGSM fails to secure model robustness, all our methods can reliably address the catastrophic overfitting issue on CIFAR-100.

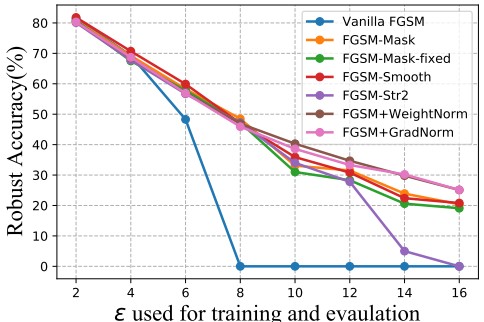

Figure 8: The robust accuracy of different methods when $\epsilon$ (perturbation size) increases.

| Methods | Clean | PGD-50-10 |
|---|---|---|
| Vanilla FGSM | 46.7% | 0% |
| FGSM-Mask | 56.4% | 25.3% |
| FGSM-Mask-Fixed | 52.3% | 22.4% |
| FGSM-Smooth | 51.6% | 25.7% |
| FGSM-Str2 | 57.9% | 25.2% |
| FGSM-GradNorm | 51.6% | 22.3% |
| FGSM-WeightNorm | 52.8% | 23.1% |

Table 7: Model performances on CIFAR-100.

## 5 RELATED WORK

**Adversarial training.** Adversarial training is one of the most effective strategies to defend against adversarial threats to machine learning systems. The idea of adversarial training origins in (Goodfellow et al., 2015) who proposes to combine clean samples and adversarial examples to train the model. (Madry et al., 2018) reformulate adversarial training as a min-max optimization problem and proposes the PGD adversarial attack, inspiring a set of follow-ups. (Zhang et al., 2019) apply a regularization term to achieve the balance between robustness and clean performance. (Shafahi et al., 2019) reduce the high cost of adversarial training by recycling the gradient information. Recent works also summarise the tricks of AT. For example, (Pang et al., 2021) reports the optimal hyperparameters for PGD-AT; (Gowal et al., 2020) explore the limits of adversarial training on CIFAR-10. In this paper, we focus on FGSM-AT, which is computationally cheap but has a severe overfitting issue, and propose a bag of tricks to fix it.

**Catastrophic overfitting.** (Wong et al., 2020) find, for FGSM-AT, its robust accuracy against PGD attacker will drop to zero after several training epochs, naming this phenomenon as catastrophic overfitting. (Rice et al., 2020) argue that catastrophic overfitting is a special case that only exists in FGSM-AT; moreover, they identify the key reason is that a weak adversarial attacker (*e.g.*, FGSM) is used in training. (Kim et al., 2021) find the decision boundary distortion is closely related to the catastrophic overfitting, and further propose to apply various step sizes for each image to address this issue. In this paper, rather than modifying adversarial attackers, we show that a pure FGSM can enable robust learners.

## 6 CONCLUSION

This work studies the solutions to the catastrophic overfitting problem in FGSM-AT from three different perspectives: Data Initialization, Network Structure, and Optimization. Comprehensive results on multiple robustness settings demonstrate that the proposed methods effectively stabilize FGSM-AT. We hope this study could encourage the community to rethink the value of FGSM-AT and potentially contribute to an elegant version of fully stabilized FGSM-AT in the future.

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
