# OpenReview forum: "Bag of Tricks for FGSM Adversarial Training "
_ICLR.cc/2023/Conference — Submitted to ICLR 2023_

### Official Review · Reviewer_sP57 · 2022-10-18

**Confidence:** 5
**Correctness:** 2
**Technical Novelty And Significance:** 1
**Empirical Novelty And Significance:** 2
**Recommendation:** 3

**Clarity, Quality, Novelty And Reproducibility:**

Clarity and quality: This paper is generally well-written and easy to follow.

Novelty: this is the major concern of this paper. The author does not provide any analyses, explanations or intuition to convince me the "bag of tricks" is generally applicable.

Reproducibility: As a pure empirical work, the authors should release the code and the checkpoint model for reproduction.


**Strength And Weaknesses:**

Strength:

1. Some observations, such as random masking and increased convolutional strides, are interesting and could be useful for practitioners.
2. Many experiments and ablation study are conducted to support the authors' claims.

Weakness:

1. My major concern is the novelty, the contribution is actually incremental. This is a pure empirical work, and the authors provide almost no explanation or intuitions about why these tricks work. This makes me doubt if these technique can be applied to more complicated tasks, such as ImageNet.

2. Although the tricks proposed can get rid of catastrophic overfitting, but the final results are not very competitive. For example, on the robustBench, "Efficient Robust Training via Backward Smoothing" can achieve 51.12% and 26.94% robust accuracy under AA on CIFAR10 and CIFAR100, respectively.

3. In order to achieve the best results, the authors should apply combinations of several tricks on WideResNet-34-10 as well, since the result of a single trick on WideResNet-34-10 is still worse than when we apply several to PreActResNet-18.

4. As mentioned in the beginning of Section 3, the authors do not use the label smoothing. What is the reason behind this? As fas as I know, adaptive label like the one in self adaptive training (https://proceedings.neurips.cc/paper/2020/file/e0ab531ec312161511493b002f9be2ee-Paper.pdf) can indeed improve the performance with little overhead.

Minor:

1. Variance should be reported for Table 2.

2. More values of $\alpha$ should be included in Figure 5(b).





**Summary Of The Paper:**

This paper comprehensively study how to mitigate or get rid of catastrophic overfitting from perspectives: data initialisation, network architecture and optimisation, and thus provides a bag of tricks for how to improve and stabilise fast adversarial training.
Specifically, the authors find that randomly masking out some input features, using smooth activation functions, increase the convolutional strides and regularising the weights on the first convolutional layer can help.

**Summary Of The Review:**

This paper empirically study the tricks to improve fast adversarial training.
Although it can provide some useful tricks for practitioner, the paper still does not reach the bar of publishing because of the concerns above.

I suggest the authors provide more analyses, explanations and intuitions, as well as the open-source code to make the findings of this paper more convincing.

---

> ### Author Response · Authors · 2022-11-19
> **To Reviewer sP57**
>
> We thank the reviewer for the detailed comments and recognition of our work. We address the concerns as follows:
>
>
> **Q1: Novelty of this work**
>
> As stated in the general response, our view is that extensively searching for multiple possible solutions to the catastrophic overfitting issue itself is new and valuable. In addition, we provide strong evidence to challenge the prior belief that randomness is the key factor in stabilizing adversarial training. For example, our methods (except FGSM-Mask) do not introduce any randomness into FGSM-AT but can still reliably secure model robustness.
>
> We additionally perform experiments on Tiny-ImageNet and showed results in the below table. Interestingly, we note catastrophic overfitting does not happen and conjecture the possible reason is that Tiny-ImageNet is a harder classification task to overfit. Nevertheless, our methods can still effectively improve robust accuracy over the vanilla FGSM-AT.
>
> | Method | PGD-50-10 |
> | :-----| ----: |
> |Vanilla FGSM|20.9%|
> |FGSM-Mask|20.6%|
> |FGSM-Mask-Fixed|20.3%|
> |FGSM-Smooth|21.9%|
> |FGSM-Str2|21.2%|
> |FGSM-GradNorm|21.2%|
> |FGSM-WeightNorm|21.3%|
>
> **Q2: Competitive performance**
>
> Thanks for the suggestion. Since FGSM is a single-step adversarial training method, it is expected to be outperformed by more advanced methods like a multi-step PGD. Nonetheless, we argue that such a single step is still valuable as it is much more computationally efficient than multi-step methods (therefore allowing further scaling up). The main goal of this paper is to show that a pure FGSM-AT can be a robust learner, free of catastrophic overfitting. We will make it clear in the next version.
>
>
>
> **Q3: The combination of tricks for WRN-34**
>
> For WRN-34, we can achieve the best robust accuracy of 50.9% by Smooth + Str2. We note this 50.9% robust accuracy is ~2% higher than any single trick in Table 6. We will add this result in the next version.
>
> **Q4: Label smoothing**
>
> Thanks for such a golden suggestion! We added label smoothing to methods in Table 5 of the main paper. We follow the setting in [1] and set the label smoothing (LS) value as 0.3. As shown in the table below, label smoothing could improve robustness further:
>
> | Method | PGD-50-10 |
> | :-----| ----: |
> |Mask+Str2|47.3%|
> |Mask+Str2+LS|48.9%|
> |Mask+Smooth|49.7%|
> |Mask+Smooth+LS|50.6%|
> |Mask+WeightNorm|46.9%|
> |Mask+WeightNorm+LS|48.6%|
> |Str2+Smooth|49.8%|
> |Str2+Smooth+LS|50.9%|
> |Str2+Smooth+WeightNorm|49.6%|
> |Str2+Smooth+WeightNorm+LS|50.7%|
>
> **Q5: More values for the Figure 5(softplus)**
>
> We plot the robust accuracy curves (https://ibb.co/9rb0wtM) for FGSM-AT + Softplus, where the alpha is ranging from 1 to 10. We note catastrophic overfitting can be effectively prevented by setting alpha <= 3. We will add this discussion in the next version.
>
>
> **Q6: Variance for Table 2 in the paper.**
>
> Thanks for the suggestion. The variance of Table 2 is reported in the below table.
>
> | Randomized Mask Ratio | PGD-50-10 |
> | :-----| ----: |
> |0-20%|0%±0%|
> |30%|48.3%±0.12%|
> |40%|47.8%± 0.16%|
> |50%|47.5%± 0.15%|
>
> | Fixed  Mask Ratio | PGD-50-10 |
> | :-----| ----: |
> |0-30%|0%±0%|
> |40%|47.4%± 0.15%|
> |50%|47.3%±0.11%|
>
> [1] Pang, Tianyu et al. “Bag of Tricks for Adversarial Training.” ArXiv abs/2010.00467 (2021): n. pag.

---

> > ### Author Response · Authors · 2022-11-24
> > **follow up**
> >
> > Dear Reviewer sP57,
> >
> > We thank you again for the valuable comments!! Could you please let us know if our rebuttal addresses your concerns? If yes, would you like to raise the score?
> >
> > Thanks
> > Authors

---

### Official Review · Reviewer_LqK6 · 2022-10-24

**Confidence:** 5
**Clarity, Quality, Novelty And Reproducibility:** The evaluation is not sufficient, but…
**Correctness:** 3
**Technical Novelty And Significance:** 3
**Empirical Novelty And Significance:** 4
**Recommendation:** 5

**Strength And Weaknesses:**

I think this work is a valuable contribution to the research on FGSM adversarial training, but I have a few concerns regarding of the current version.

1. I think there is a slight discrepancy between the title and the core idea of the paper. Throughout the whole paper, the first glance of three tricks is to overcome catastrophic overfitting in FGSM-AT. Their improvement in performance is not very noticeable. Moreover, I think if the title is ``bags of tricks``, the studies of hyperparameter and other perspectives are also considered. Maybe the title like ``three things everyone should know ...`` is better.

2. In Figure 2, the authors show the results by using a masking trick. When mask ration is equal to 30%, why does the training experience catastrophic overfilling after the first learning rate drops? My concern for the Mask-Fixed scheme is whether the result is related to random initialization. For instance, we just achieved a good random mask that is beneficial to FGSM-AT. So I think it's better to run the experiments multiple times and report the mean and variance score. Masking techniques are similar to random erasing [1] and cutout [2]. How well does it work using this classic data regularization technique?

3. In section 3.2, they studied the impact of the network structure. Some related work is missing. In [3], the authors "modernize" a standard ResNet toward the design of a vision transformer, which shows promising robustness behaviors. And the work in [4] uses the NAS technique with different modules to improve the model's robustness.

4. The stride of CNN is actually related to the spatial downsampling. Recently, DiffStride [5] is a pooling layer with learnable strides. I suggest the authors try this adaptive module. Moreover, I think this modification of strides is linked to the frequency model in [6], which also modified the first layer in CNN. Moreover, I am interested in the impact of the large kernel used in recent CNN [3, 7].

5. I think it's better to report the training time in all experiments, especially for the tables.

6. In Table 5, what will the results be when we combine multiple tricks? For example, we consider: Mask-fixed+Str2 (+Weightnorm), and Mask+Smooth+WeightNorm. A similar question is also for Table 6. What will the results be when considering Smooth+Str2, mask+smooth, and Smooth+Str2+WeightNorm?

7. In Table 7 and Figure 8, it's better to show the results of a combination of different tricks.

8. A recent FGSM-AT [8] is missing.

9. FGSM-AT is an effective adversarial training that can train deep model on large-scale dataset, like ImageNet. So it's better to evaluate the mentioned tricks on a large dataset, like Tiny-ImageNet or ImageNet.

[1] Random Erasing Data Augmentation. AAAI 2020.

[2] Improved Regularization of Convolutional Neural Networks with Cutout. Arxiv 2017.

[3] A ConvNet for the 2020s, CVPR 2022.

[4] Anti-Bandit Neural Architecture Search for Model Defense. ECCV 2020.

[5] DiffStride: Learning strides in convolutional neural networks. ICLR 2022.

[6] High-frequency component helps explain the generalization of convolutional neural networks. CVPR 2020.

[7] Scaling up your kernels to 31x31: Revisiting large kernel design in cnns. CVPR 2022.

[8] Revisiting and advancing fast adversarial training through the lens of bi-level optimization. ICML 2022.

**Summary Of The Paper:**

This paper systematically studies different tricks for FGSM adversarial training (FGSM-AT) on CIFAR-10/100. They found that there are three simple tricks that overcome the catastrophic overfitting in FGSM-AT: Data Initialization, Network Structure, and Optimization.

**Summary Of The Review:**

This paper systematically studies different tricks for FGSM-AT, but its main target is to overcome the catastrophic overfitting. If the authors consider doing a topic on ''bags of tricks," I think the authors should give more takeaways and put forward a standard baseline setting for future work. I have many concerns about the current version. For more details, please refer to the [Strength And Weaknesses].

---

> ### Author Response · Authors · 2022-11-19
> **To Reviewer LqK6 (1/2)**
>
> We thank the reviewer for the detailed comments and recognition of our work. We address the concerns as follows:
>
> **Q1: Discrepancy between the title and the core idea of the paper**
>
> A: Thanks for your suggestion. We will consider the following two titles, “Bag of Tricks of Stabling FGSM Adversarial Training” and “Three Things You Should Know for Stabling FGSM Adversarial Training”, for better reflecting the major contributions of this paper.
>
>
>
> **Q2: Overfitting when the learning rate decays**
>
> The previous work [1] found that a small learning rate would increase the overfitting risk. Therefore, when the learning rate decays, catastrophic overfitting happens easily. We will discuss this reference accordingly in the next version.
>
> **Q3: Various random seeds for mask methods**
>
> In Table1, the results are already averaged over three random seeds. Also, please see our response to the reviewer sPmb about 10 random seed experiments. Under all these settings, we note that our methods can reliability bring in consistent robustness improvements.
>
>
> **Q4: Random erasing and Cutout**
>
> For RandomErase, we experiment with various probabilities and show their robust accuracy in the figure(https://ibb.co/QJH8kWy). The robust accuracy of Cutout is shown in the figure(https://ibb.co/sb9r7NZ). Compared to vanilla training, these two methods can delay catastrophic overfitting, but they fail to stop catastrophic overfitting, i.e., the final robust accuracy is zero. We will add these discussions in the next version.
>
> **Q5: Other combinations of tricks**
>
> In Table 5, we combine multiple tricks to see if we can get better results. Given Mask-Fixed does not show better results than Mask, we do not combine Mask-Fixed with other methods in our paper. The robust accuracy of Mask-Fixed + Smooth is 48.3%, lower than Mask + Smooth’s 49.7%.  In Table 6 (WideResNet), the best robust accuracy 50.9% is achieved by Smooth + Str2 after we try different combinations. Note this 50.9% robust accuracy is ~2% higher than any single trick in Table 6. We will add and discuss these results accordingly in the next version.
>
>
> **Q6: FGSM-AT with DiffStride[4]**
>
> Thanks for the suggestion.  We follow the PyTorch version of DiffStride (https://github.com/annotated-paper-reviews/diffstride). Experiments (https://ibb.co/WnyPKJ5)  show that DiffStride fails to mitigate catastrophic overfitting of FGSM-AT. We will add this discussion in the next version.
>
> **Q7: NAS structure**
>
> Thanks for the suggestion. Following the PyTorch implementation (https://github.com/bczhangbczhang/ABanditNAS), We test the network in [2] and show robust performance in this figure (https://ibb.co/Y3RRy0N ). The searched network could delay catastrophic overfitting but fail to stop it. The robust accuracy drops to zero after 23 training epochs. We will add this discussion in the next version.
>
>
> **Q8: Training time**
>
> Please see the table below. In short, most of our methods do not bring in additional computation overheads to the vanilla FGSM-AT.
> | Method | Time(s/epoch) |
> | :-----| ----: |
> |Vanilla FGSM|40|
> |FGSM-Mask|40|
> |FGSM-Mask-Fixed|40|
> |FGSM-Smooth|43|
> |FGSM-Str2|37|
> |FGSM-GradNorm|109|
> |FGSM-WeightNorm|42|

---

> > ### Author Response · Authors · 2022-11-24
> > **follow up**
> >
> > Dear Reviewer LqK6,
> >
> > We thank you again for the valuable comments!! Could you please let us know if our rebuttal addresses your concerns? If yes, would you like to raise the score?
> >
> > Thanks
> > Authors

---

> ### Author Response · Authors · 2022-11-19
> **To Reviewer LqK6 (2/2)**
>
> **Q9: The combination of tricks for Figure 8 and Table 7**
>
> For Figure 8, we show the performance of the combination of our methods in this figure (https://ibb.co/mRbQgMb), where “Ours” represents our methods. For Table 7, we can get a better result of 28.4% by combining FGSM-Str2 and FGSM-Smooth, which is ~3% higher than that of any single trick in Table 7. We will add these results in the next version.
>
>
> **Q10: Missing the work [3]**
>
> Thanks for the suggestion. We run the Fast-BAT, and the robust accuracy on CIFAR-10 under PGD-50 attack is 48.70%.  Our methods attain comparable performance but are more efficient than Fast-BAT. We will discuss this comparison accordingly in the next version.
>
> **Q11: Large datasets**
>
> Thanks for the suggestion. We test our methods on Tiny-ImageNet with PreActResNet-18. The training recipe is the same as that in the paper. Interestingly, we did not observe catastrophic overfitting during the training. We conjecture this is because Tiny-ImageNet classification is more challenging and difficult to overfit. Adding our methods to vanilla FGSM-AT can lead to further robustness improvements, as shown in the below table.
>
> | Method | PGD-50-10 |
> | :-----| ----: |
> |Vanilla FGSM|20.9%|
> |FGSM-Mask|20.6%|
> |FGSM-Mask-Fixed|20.3%|
> |FGSM-Smooth|21.9%|
> |FGSM-Str2|21.2%|
> |FGSM-GradNorm|21.2%|
> |FGSM-WeightNorm|21.3%|
>
>
> **Q12: The motivation and the goal of this work**
>
> Thanks for the suggestion. The tricks in the paper aim at preventing catastrophic overfitting in FGSM-AT, rather than pursuing the SOTA of FGSM-AT. The standard baseline in this work is the vanilla FGSM-AT, whose robustness drops to zero during training. Comprehensive experiments show that our tricks can prevent overfitting and improve the baseline. We will make it more clear in the next version.
>
> [1] Smith, Leslie N. and Nicholay Topin. “Super-convergence: very fast training of neural networks using large learning rates.” Defense + Commercial Sensing (2019).
>
> [2] Chen, Hanlin et al. “Anti-Bandit Neural Architecture Search for Model Defense.” ArXiv abs/2008.00698 (2020): n. pag.
>
> [3] Zhang, Yihua et al. “Revisiting and Advancing Fast Adversarial Training Through The Lens of Bi-Level Optimization.” ArXiv abs/2112.12376 (2022): n. Pag.
>
> [4] Riad, Rachid et al. “Learning strides in convolutional neural networks.” ArXiv abs/2202.01653 (2022): n. pag.

---

### Official Review · Reviewer_sPmb · 2022-10-26

**Confidence:** 5
**Correctness:** 4
**Technical Novelty And Significance:** 2
**Empirical Novelty And Significance:** 3
**Recommendation:** 6

**Clarity, Quality, Novelty And Reproducibility:**

The paper is well written, and presents its key ideas in a lucid manner. While the technical contributions are indeed limited, the empirical set of experiments presented do serve as a useful guide for modifications in training (such as setting stride=2 instead of 1 etc.).

**Strength And Weaknesses:**

Strengths:
1) The paper is well-written, and motivates each aspect considered in a clear manner.
2) The three primary modifications identified are seen to be fairly effective in stabilizing FGSM-based adversarial training, while not requiring any large additions in the computational footprint needed for training.
3) Moreover, the methods themselves, such as masking out a fraction of the input pixels, or using larger stride for convolutions and smooth activation functions are fairly simple, and thus will likely be quite of high utility in practice.
4) The paper also presents a fairly thorough ablative study to better understand the relative contribution of these different techniques. In particular, the results obtained on the WideResNet-34-10 indicate that the findings are not overly specific to one network-type alone, and yield robust networks in sharp contrast to that of Fast-FGSM-AT.
5) In addition, the results presented over larger perturbation radii also help establish the validity and generality of the techniques proposed to mitigate catastrophic failure during FGSM training, and are seen to be effective even at 16/255 for L-infinity based adversaries.



Weaknesses:
1) The paper slightly lacks from the viewpoint of technical novelty, but the empirical findings presented are still useful for the community at large. For example, the notion of using a fixed mask to help stabilize training seems to suggest that reducing the dimensionality of the input is effective, and this might correlate well with the “effective” strength of an L-infinity based adversary in being able to perturb fewer pixels, though the epsilon-budget stays fixed.
2) Moreover, the paper could have been significantly improved by including discussions on the L2 threat model with FGSM-AT, to yield a better understanding of the techniques, and the extent of their generality.
3) Given that the main theme of the paper is about improving the stability of single-step FGSM training, a more comprehensive experiment set would be expected over a large statistical sample - the average over three random seeds alone is perhaps a bit lacking. Given that the models can be trained extremely quickly, the paper could perhaps have shown results over 10 random seeds, given that even FGSM-AT behaves quite differently at times with different initializations. This certainly is not required for all the different values presented in all the tables, but could have been included for final comparisons.
4) The ablation with WideResNet-34-10 is a fairly pertinent section of the paper, since it presents an important leap in stability over Fast-AT. For example, were these models too trained for 110 epochs similar to that of results present for the ResNet models, alongside other factors such as learning rate? Other observations from Table-6, such as that of the stride being set to 2 appearing to be the single-most effective factor could have been discussed in more detail, and is fairly important to assess the relative importance amongst other proposed ingredient changes.
5) Furthermore, since robust overfitting is observed at a much larger scale in these settings, validation-based early stopping might help present potential improvements as well, and would better set apart the improvements achieved over Fast-AT which uses early stopping necessarily. Similarly, more detailed comparisons could have been made with respect to GradAlign in Figure-8 with larger perturbation radii for the L-infinity threat model. Overall, the contributions of the paper can perhaps be strengthened significantly by including a more detailed and broadened discussion of the ablations section, with other matter potentially moved to the appendix.
6) Could the authors also comment on the low clean accuracy achieved by several models that utilize the smooth activation function, such as those presented in Table-3,6? Since the tradeoff between robustness and accuracy is well known, focusing on robust accuracy in a vacuum may not be entirely indicative of the phenomenon at hand. For example, is it believed that these networks overfit to a far smaller extent, and have consequent impacts on the level of robustness proportionately achieved?


**Summary Of The Paper:**

In this work, the authors study the phenomenon of catastrophic failure during Fast Gradient Sign Method (FGSM) adversarial training, and identify a collection of techniques that help stabilize the training process. In particular, the paper proposes three main ingredients: (1) modifying data wherein a random/fixed subset of input pixels are masked to zero, (2) the use of larger strides for convolutions and smooth activation functions, and (3) regularization of weights or gradients.

**Summary Of The Review:**

Though the technical contributions are limited, the paper presents a fairly simple set of modifications to Fast-FGSM-AT with minimal computational overhead. The overarching guidelines, such as using a larger stride for convolutions etc are useful in practice, and is a good contribution overall. As mentioned in the weaknesses, results over a larger number of seeds could be presented to better establish the generality and validity of results, alongside a more detailed discussion for pertinent sections such as the results obtained on WideResNet, larger perturbation radii etc. I would be willing to raise my score further if these issues could be addressed.

---

> ### Author Response · Authors · 2022-11-19
> **To Reviewer sPmb**
>
> We thank the reviewer for the detailed comments and recognition of our work. We address the concerns as follows:
>
> **Q1: Lack Technical Novelty**
>
> As stated in the general response, our view is that extensively searching for multiple possible solutions to the catastrophic overfitting issue itself is new and valuable. In addition, we provide strong evidence to challenge the prior belief that randomness is the key factor in stabilizing adversarial training. For example, our methods (except FGSM-Mask) do not introduce any randomness into FGSM-AT but can still reliably secure model robustness.
>
> **Q2: Discussion about L2-threat model with FGSM-AT**
>
> Thanks for the suggestion. We implement the vanilla L2 FGSM-AT with the same training recipe in our manuscript. We set the epsilon as 128/255 following the setting of [1]. Interestingly, we do NOT observe catastrophic overfitting along the training, as shown in the figure (https://ibb.co/0jnNsv3). In our future work, we will continue exploring the relationship between catastrophic overfitting and different attackers.
>
> **Q3: Results over 10 random seeds**
>
> Thanks for the suggestion. We report the experimental results with 10 random seeds in the table below. We note the variation of the robustness against the PGD-50 attack and AutoAttack is small. We will add these results in the next version.
>
> | Method | PGD-50-10 |AA|
> | :-----:| ----: | :----: |
> |Vanilla FGSM|0%|0%|
> |FGSM-Mask|48.3%±0.12%|44.1%±0.13%|
> |FGSM-Mask-Fixed|47.3%±0.11%|42.5%±0.3%|
> |FGSM-Smooth|47.9%±0.40%|43.1%±0.22%|
> |FGSM-Str2|48.4%±0.35%|44.4%±0.22%|
> |FGSM-GradNorm|46.4%±0.24%|42.6%±0.15%|
> |FGSM-WeightNorm|46.9%±0.50%|42.9%±0.31%|
>
>
>
>
> **Q4: Training recipe for WideResNet training**
>
> The training recipe for WideResNet is exactly the same as that of PreActResNet, including the initial learning rate (0.1), total epochs (110), and weight decay (5e-4). We will make it clear in the next version.
>
>
>
> **Q5: The effectiveness of the large stride size**
>
> Thanks for your interest in this method. FGSM-Str2 is effective for various networks with $\epsilon$=8. By increasing the stride size from 1 to 2 or 3, the catastrophic overfitting problem is successfully addressed. As shown in this figure (https://ibb.co/y8NFL26 ), when the stride is set to 1 (i.e., the vanilla baseline), the robust accuracy quickly drops to zero. The robust accuracy curve is much more stable when the stride is set to 2 or 3. Among different stride options in our study, we find that FGSM-AT with a stride of 2 achieves the highest robust accuracy. Moreover, we conjecture that the catastrophic overfitting is possibly related to the receptive field of neural networks, and we will continue the exploration in our future works.
>
> **Q6: Validation-based early stopping**
>
> Thanks for the suggestion. We revisit the validation-based early stopping in the paper [2]. Their early stopping time is when the current robust accuracy is 20% lower than the previous epoch’s. Our experiments show that this validation-based early stopping nearly has no effects on our method.
>
>
> **Q7: Comparison with GradAlign**
>
> We compare GradAlign with our methods under PGD-50 attack with various $\epsilon$. We report the results in this figure (https://ibb.co/mRbQgMb), where  “Ours” represents the combination of our tricks. Our methods have similar results to GradAlign under attacks with different $\epsilon$. We will add this comparison in the next version.
>
> **Q8: Low accuracy brought by the smooth activation and the trade-off between clean and robust accuracy**
>
> We share the similar conjecture with the reviewer that the smoother activation function can make the network less prone to overfitting, therefore, showing such a trade-off between clean and robust accuracy. We will explore it in future works.
>
>
> **Q9: The relative importance amongst other proposed ingredient changes.**
>
> Thanks for the suggestion. We want to stress that our main point is to provide a set of “simple” solutions/tricks to alleviate the catastrophic overfitting issue in FGSM-AT. Regarding relative importance, we believe the FGSM+Str2 and FGSM+Smooth are more important as these two methods firstly prove that modifications on network structure could stop catastrophic overfitting. We hope our findings can encourage researchers to investigate model robustness from multiple perspectives and help lay the foundation of SOTA robust models in the future.
>
>
> [1] Croce, Francesco et al. “RobustBench: a standardized adversarial robustness benchmark.” ArXiv abs/2010.09670 (2021): n. Pag.
>
> [2] Wong, Eric et al. “Fast is better than free: Revisiting adversarial training.” ArXiv abs/2001.03994 (2020): n. pag.

---

> > ### Author Response · Authors · 2022-11-24
> > **follow up**
> >
> > Dear Reviewer sPmb,
> >
> > We thank you again for the valuable comments!! Could you please let us know if our rebuttal addresses your concerns? If yes, would you like to raise the score?
> >
> > Thanks
> > Authors

---

### Official Review · Reviewer_8ACT · 2022-11-20

**Confidence:** 4
**Correctness:** 3
**Technical Novelty And Significance:** 2
**Empirical Novelty And Significance:** 3
**Recommendation:** 5

**Clarity, Quality, Novelty And Reproducibility:**

The work is clearly written with good quality but the novelty is limited in the sense of lacking a good explanation. I believe that this work can be reproducible.

**Details Of Ethics Concerns:**

N.A.

**Strength And Weaknesses:**

Strength 1: This work performs a comprehensive analysis on FGSM AT

Strength 2: Competitive performance.

Weakness1: The main concern is that this work mainly relies on empirical results on showing effectiveness. For example, it is fully unclear why masking a subset of the input pixels can stabilize FGSM-AT. Why does it help prevent catastrophic overfitting? Why does catastrophic overfitting happen to ViT not CNN? Why does larger stride help? To my understanding, the core issue in FGSM AT its how to prevent catastrophic overfitting, a recent work [1] shows random noise augmentation is sufficient for preventing catastrophic overfitting and the reason of its success is analyzed in another work [2]. Can Non-robust feature perspective in [2] be used to explain to the success of the proposed techniques in this work? Or do the authors have their own new perspective? From my understanding, to provide explanations for why the techniques can be more important than the tricks themselves, especially in the adversarial ML community. Otherwise, the takeaway from this work can be limited.

Weakness2: The technical contributions might also be limited.


[1] Fast Adversarial Training with Noise Augmentation: A Unified Perspective on RandStart and GradAlign, https://arxiv.org/pdf/2202.05488.pdf

[2] Understanding Catastrophic Overfitting in Fast Adversarial Training From a Non-robust Feature Perspective, https://openreview.net/forum?id=UO8UP_xDMwD

**Summary Of The Paper:**

As the title suggests, this work studies bags of tricks for FGSM AT.


**Summary Of The Review:**

Overall, I believe this work has some strong empirical findings but lacks a good explanation of why they help prevent catastrophic overfitting. The limited technical contribution might also be a concern. I decide to submit this late review considering it only has three reviews, however, the authors do not need to address my concerns considering that my review is late.

Even though my score is 5 (marginally below the acceptance threshold), I am totally okay if this paper is accepted.

---

> ### Author Response · Authors · 2022-11-22
> **To reviewer 8ACT**
>
> We thank the reviewer for the detailed comments and the appreciation of our work. We address the concerns as follows:
>
> **Q1: Lacking in-depth explanations**
>
> As stated in our general response, we would like to highlight that, rather than providing an in-depth analysis of why one specific method works (as the related works [2][3]), the main position of this paper is to extensively show the possibility of addressing the catastrophic overfitting problem from many different _**(and previously unknown)**_ perspectives. For example, while previous works generally believe that _**“random noise”**_ is the key factor to preventing catastrophic overfitting in FGSM-AT, this paper alternatively shows that _**neither “random” nor “noise”**_ is a necessary component to ensure the success of FGSM-AT. We believe offering these new perspectives (though empirical) on tackling catastrophic overfitting is still valuable and worthy to be known to the general research community, and more essentially, can motivate (both empirical and theoretical) future work along this research direction.
>
>
> **Q2: Using [2] to help understand our tricks.**
>
> Thanks for pointing out this related work [2]. Following the setup of Figure 7 in [2] (https://ibb.co/4sVbWDv), we leverage this tool to understand our FGSM-Mask, i.e., replacing the noise augmentation with masking a subset of pixels in this experiment. We report the result in the figure (https://ibb.co/TwG9qXV). Similarly, we note the $\nabla_{FGSM}$ of CO drops significantly. This observation suggests that the non-robust feature perspective in [2] can help explain our FGSM-Mask. We will add the discussions accordingly in the next version.
>
> While one limitation of [2] is that it cannot be used to explain our other tricks, including FGSM-Str2 and FGSM-WeightNorm. This is because the explanation mechanism of [2] is built upon controlling input types (e.g., either clean or adversarial data), which therefore cannot be (trivially) adapted for explaining methods with changes of architecture or regularization terms. We leave it as a possible future direction for comprehensively understanding and explaining our methods.
>
>
> [1] Wong, Eric et al. “Fast is better than free: Revisiting adversarial training.” ArXiv abs/2001.03994 (2020): n. pag.
>
> [2] Understanding Catastrophic Overfitting in Fast Adversarial Training From a Non-robust Feature Perspective, https://openreview.net/forum?id=UO8UP_xDMwD
>
> [3] Niu, Axi et al. “Fast Adversarial Training with Noise Augmentation: A Unified Perspective on RandStart and GradAlign.” (2022).

---

> > ### Author Response · Authors · 2022-11-24
> > **follow up**
> >
> > Dear Reviewer 8ACT,
> >
> > We thank you again for the valuable comments!! Could you please let us know if our rebuttal addresses your concerns? If yes, would you like to raise the score?
> >
> > Thanks
> > Authors

---

### Author Response · Authors · 2022-11-19
**General Response**

We appreciate all the thoughtful feedback from reviewers, which will greatly help us improve this manuscript. We also thank the reviewers for recognizing the value of our strategies for mitigating catastrophic overfitting.

As suggested by the reviewers, we included the following experimental results in this rebuttal:

1. Statistics: We repeat major experiments with 10 random seeds, and report robust accuracy and variances. We note our methods perform reliably with very small variance. See response to Reviewer sPmb for details.

2. Large dataset: we train our methods on a larger dataset, Tiny-ImageNet. We note our methods can consistently improve robustness. See response to Reviewer LqK6 and  sP57 for details.
3. Other structures: We conduct additional experiments with DiffStride[2] and ABanditNAS[3]; nonetheless, they cannot prevent catastrophic overfitting. See response to Reviewer LqK6 for details.
4. Other data augmentation techniques: We validate that neither RandomErasing nor CutOut can prevent catastrophic overfitting. See response to Reviewer LqK6 for detail.
5. Combination of tricks: we combine tricks on various models and datasets. Results show that combined tricks can get better results than the single one. See response to Reviewer LqK6 and  sP57 for details.
6. Training time: we report the time per epoch of each method. Note that our methods can run as fast as the vanilla FGSM-AT. See response to Reviewer sP57 for details.
7. L_2 threat: we do not observe catastrophic overfitting on FGSM-AT with L_2 threat. See response to Reviewer sPmb for details.
8. Label smoothing: label smoothing can further improve the robust accuracy of our methods. See response to Reviewer sP57 for details.
9. Validation-based early stopping: we follow the early stopping strategy in [1] and find that early stopping fails to boost robustness further. See response to Reviewer sPmb for details.
10. Various alpha for Softplus: we try Softplus with various alpha values and validate that the activation function can stop catastrophic overfitting when the alpha is less than a certain value. See response to Reviewer sP57 for details.

Lastly, we would like to reiterate that rather than providing an in-depth analysis of why one specific method works, this paper alternatively focuses on extensively showing the possibility of addressing the catastrophic overfitting problem from many different perspectives. To our best knowledge, this is the first work of such type, and we hope our work can provide a new perspective on motivating future work on comprehensively understanding and addressing catastrophic overfitting in adversarial training.

[1] Wong, Eric et al. “Fast is better than free: Revisiting adversarial training.” ArXiv abs/2001.03994 (2020): n. Pag.

[2] Riad, Rachid et al. “Learning strides in convolutional neural networks.” ArXiv abs/2202.01653 (2022): n. pag.

[3] Chen, Hanlin et al. “Anti-Bandit Neural Architecture Search for Model Defense.” ArXiv abs/2008.00698 (2020): n. pag.

---

### Decision · Program_Chairs · 2023-01-20

**Decision:**

Reject

**Justification For Why Not Higher Score:**

Although the paper considers an interesting and important practical problem, reviewers have concerns about the novelty of the paper is limited. In addition, there are some concerns about the final results not being very competitive.


**Justification For Why Not Lower Score:**

N/A

**Metareview: Summary, Strengths And Weaknesses:**

This paper considers a problem in adversarial training using FGSM: the robust accuracy abruptly drops to zero within a single training step. The paper investigates a collection of tricks that mitigate this catastrophic overfitting issue.

Although the paper considers an interesting and important practical problem, reviewers have concerns about the novelty of the paper is limited. In addition, there are some concerns about the final results not being very competitive.